# Hybrid Material Based on *Vaccinium myrtillus* L. Extract and Gold Nanoparticles Reduces Oxidative Stress and Inflammation in Hepatic Stellate Cells Exposed to TGF-β

**DOI:** 10.3390/biom13081271

**Published:** 2023-08-20

**Authors:** Mara Filip, Ioana Baldea, Luminita David, Bianca Moldovan, Gabriel Cristian Flontas, Sergiu Macavei, Dana Maria Muntean, Nicoleta Decea, Adrian Bogdan Tigu, Simona Valeria Clichici

**Affiliations:** 1Department of Physiology, ‘‘Iuliu Hatieganu’’ University of Medicine and Pharmacy, 1–3 Clinicilor Street, 400006 Cluj-Napoca, Romania; flontas.gabrielcristian@elearn.umfcluj.ro (G.C.F.); nicoleta_decea@yahoo.com (N.D.); sclichici@umfcluj.ro (S.V.C.); 2Department of Chemistry, Faculty of Chemistry and Chemical Engineering “Babes-Bolyai” University, 11 Arany Janos Street, 400028 Cluj-Napoca, Romania; luminita.david@ubbcluj.ro (L.D.); bianca.moldovan@ubbcluj.ro (B.M.); 3National Institute for Research and Development of Isotopic and Molecular Technologies, Donath Street, No. 67-103, 400293 Cluj-Napoca, Romania; sergiu.macavei@itim-cj.ro; 4Department of Pharmaceutical Technology and Biopharmaceutics, ‘‘Iuliu Hatieganu’’ University of Medicine and Pharmacy, 8 Victor Babeș Street, 400347 Cluj-Napoca, Romania; dana.muntean@umfcluj.ro; 5Medfuture Research Center for Advanced Medicine, “Iuliu Hatieganu” University of Medicine and Pharmacy, 4 Louis Pasteur Street, 400347 Cluj-Napoca, Romania; bogdan.tigu@elearn.umfcluj.ro

**Keywords:** gold nanoparticles, polyphenols, oxidative stress, inflammation, fibrosis

## Abstract

(1) Background: The study aimed to investigate the impact of gold nanoparticles capped with *Cornus sanguinea* (NPCS) and mixed with a fruit extract (*Vaccinum myrtillus* L.—VL) on human hepatic stellate cells (LX-2) exposed to TGF-β. (2) Methods: NPCS were characterized by UV-Vis, transmission electron microscopy (TEM), zeta potential measurement, X-ray diffraction (XRD) and energy dispersive spectroscopy (EDX). The cytotoxic effects of VL, NPCS and of the hybrid compounds obtained by mixing the two components in variable proportions (NPCS-VL) were assessed. LDH activity, MDA levels, secretion of inflammation markers, the expression of fibrogenesis markers and collagen I synthesis were estimated after treating the cells with a mixture of 25:25 μg/mL NPCS and VL. (3) Results: TEM analysis showed that NPCS had spherical morphology and homogenous distribution, while their formation and elemental composition were confirmed by XRD and EDX analysis. TGF-β increased cell membrane damage as well as secretion of IL-1β, IL-1α and TLR4. It also amplified the expression of α-SMA and type III collagen and induced collagen I deposition. NPCS administration reduced the inflammation caused by TGF-β and downregulated α-SMA expression. VL diminished LDH activity and the secretion of proinflammatory cytokines. The NPCS-VL mixture maintained IL-1β, IL-1α, TLR4 and LDH at low levels after TGF-β exposure, but it enhanced collagen III expression. (4) Conclusions: The mixture of NPCS and VL improved cell membrane damage and inflammation triggered by TGF-β and mitigated collagen I deposition, but it increased the expression of collagen III, suggestive of a fibrogenetic effect of the hybrid material.

## 1. Introduction

Chronic liver diseases (CLD) represent a serious public health issue worldwide due to their high prevalence (currently an estimated 1.5 billion patients) and to the lack of curative treatment options [1]. They are among the leading causes of morbidity and mortality globally, responsible for approximately two million deaths yearly [2].

The etiology and progression of CLD depend on the presence of certain factors, such as chronic infection with hepatitis B virus (HBV) or hepatitis C virus (HCV), alcoholic steatohepatitis (ASH), non-alcoholic steatohepatitis (NASH), genetic diseases and autoimmunity [3]. Regardless of their cause, the evolution of CLD is similar, with liver fibrosis, still reversible, followed by cirrhosis, an irreversible stage of disease. In evolution, liver cirrhosis leads to hepatic insufficiency and is a major risk factor for hepatocellular carcinoma, a primary liver malignancy that accounts for approximately 700,000 deaths yearly [4,5].

In what concerns the underlying pathophysiological mechanisms, the activated hepatic stellate cells (HSC) play a central role in fibrosis [6]. HSC are located in the perisinusoidal space of Dissé and, together with the hepatocytes, the macrophages and the sinusoidal endothelial cells, they form the structure of the liver. In healthy individuals, HSC store vitamin A and retinoids and synthetize a limited amount of extracellular matrix (ECM) proteins [7]. In the presence of chronic hepatic injury, HSC undergoes a process of activation and transdifferentiation into myofibroblasts [8]. These cells proliferate intensely, produce an increased amount of ECM proteins, such as type I collagen and type III collagen [6] and also induce actin cytoskeleton disorganization. As a result, the liver parenchyma is replaced with fibrous connective tissue which, along with the tonic contraction of the activated HSC, increases the liver stiffness; this is characteristic of fibrosis and cirrhosis [9].

The activation of the HSC can be initiated by the immune cells in the liver, especially by the macrophages [7]. They recognize pathogen-associated molecular patterns (PAMPs) and damage-associated molecular patterns (DAMPs) released in liver injury. This results in inflammasome activation in the macrophages and release of pro-inflammatory cytokines and chemokines, which, in turn, can activate other HSC and amplify fibrosis [10]. Additionally, DAMPs can activate HSC through Toll-like receptors (TLRs), which can stimulate macrophage chemotaxis and induce cytokine secretion, sensitizing HSC to TGF-β [11,12]. HSC can also be activated by reactive oxygen species (ROS) or directly by certain cytokines. One of their most potent activators is TGF-β, with TGF-β1 being the most investigated isoform involved in liver fibrogenesis [13]. Physiologically, this cytokine is stored in its inactive form. When activated, TGF-β initiates a signaling pathway through Smad proteins, causing the transdifferentiation of HSC into collagen-producing myofibroblasts and the inhibition of hepatocyte production [12].

Conventional therapy has limited efficiency in CLD, as it is not capable of providing an appropriate concentration of the active substance into the target structures. Nanotechnology has proved to be an effective solution to this problem, as it can be used to directly introduce therapeutic agents into the injured areas [14]. Nanoparticles (NPs) are versatile molecules which have been extensively used in biomedicine due to their biocompatibility, stability and lack of toxicity. They can be easily functionalized for targeted delivery of medication [3,15]. The beneficial effect of gold NPs was demonstrated on experimental models of liver fibrosis: in combination with Silymarin, they ameliorate CCl_4_-induced fibrosis [16] and improve alcohol and methamphetamine-induced hepatic lesions [17].

Based on these data, our study aimed to test the effects of a hybrid material, gold nanoparticles capped with polyphenols from *Cornus sanguinea* L. extract (NPCS) mixed with blueberry fruit extract (VL), on HSC exposed to TGF-β. *Cornus sanguinea* L. extract was chosen due to its high content of bioactive compounds and its good reducing and antioxidant capacity [18,19], while blueberry fruit extract has proven antimutagenic, antitumor, antiviral and antioxidant properties [20]. Additionally, blueberry fruit extract has demonstrated a strong anti-inflammatory effect in liver disease due to its ability to interfere with the metabolism of arachidonic acid, its antioxidant effects and its inhibitory action on the accumulation of phagocytic cells at the site of injury [21,22]. The effect of the hybrid material (NPCS-VL) on LX-2 cells was evaluated by cell viability quantification; assessment of oxidative stress, inflammation and cell membrane damage; and the expression of fibrogenesis markers.

## 2. Materials and Methods

### 2.1. Vegetal Material and Reagents

The blueberries were bought from the Central Market in Cluj-Napoca and the common dogwood fruits were collected from a forest in Alba County. All fruits were frozen until use. To obtain the fruit extract, the fruit puree (5 g) was mixed with distilled water (100 mL), stirred 1 h at 25 °C and then vacuum-filtered. The Folin–Ciocalteu assay [23] was applied to determine the total phenolics from the fruit extract by mixing 0.25 mL of fruit extract with 1.5 mL of Folin–Ciocalteu reagent and, after 5 min, adding 1.2 mL of 0.7 M Na_2_CO_3_ solution. After 2 h in the dark, at 25 °C, the solution was spectrophotometrically investigated and its absorbance relative to a blank sample was recorded at 765 nm. A calibration curve was used to express the result in µg gallic acid equivalents (GAE)/mL extract.

### 2.2. Synthesis of NPCS Nanoparticles and Their Characterization

The synthesis of NPCS was achieved by the reduction reaction of gold ions from tetrachloroauric acid with bioactive compounds from the fruit extract, which act both as reducing and capping agents [18]. For this purpose, 20 mL fruit extract was brought to pH = 7.5 using a 0.1 M NaOH solution and then added dropwise over 100 mL boiling 1 mM solution of HAuCl_4_. The resulting mixture was stirred (without heating) for 30 min, during which the formation of colloidal gold was observed by the change in the color of the solution from light cherry to purple. The gold colloidal solution was centrifuged for 30 min at 12,000 rpm, the supernatant was decanted and the residue was washed 2 times with bidistilled water. The resulting gold pellet was resuspended in 5 mL distilled water and the gold content of the obtained colloidal solution was evaluated by atomic absorption spectrometry using an AVANTA PM atomic absorption spectrometer (GBC Scientific Equipment, Braeside, Australia) equipped with a graphite furnace (GF 3000) and programmable autosampler (PAL 3000, GBC Scientific Equipment).

The obtained gold nanoparticles were characterized spectrophotometrically (using a Perkin Lambda 25 double beam spectrophotometer) and microscopically (using a H-7650 120 kV Automatic transmission electron microscope, Hitachi, Tokyo, Japan). Moreover, their hydrodynamic diameter and their zeta potential were measured (using a Zetasizer Nano ZS-90 instrument, Malvern Instruments Ltd., Malvern, UK). X-ray diffraction (XRD) was performed, as previously published [24], using a Smart Lab Rigaku diffractometer with a graphite monochromator with Cu-Ka radiation (k ¼ 1:54 Å); X-ray source: Anode Cu, 9 kW at room temperature. Additionally, the morphology of NPCS was investigated by Field-Emission Scanning Electron Microscopy combined with energy dispersive X-ray spectroscopy (EDX) (HITACHI HD-2700, Cold field emission electron beam accelerated at 200 kV and 10 μA, Magnification 100×–10,000,000×).

### 2.3. Biological Assays

#### 2.3.1. Reagents

2-thiobarbituric acid, TGF-β, Bradford reagent and gold chloride solution were obtained from Sigma-Aldrich Chemicals GmbH (Darmstadt, Germany). The antibodies against alpha-smooth muscle actin (α-SMA), type I collagen, type III collagen and β-actin were obtained from Santa Cruz Biotechnology (Santa Cruz, CA, USA). An antibody against collagen I conjugated with Alexa fluor 455 was used for fluorescence visualization of collagen I. ELISA kits for interleukin-1β (IL-1β), interleukin-1α (IL-1α) and Toll-like receptor 4 (TLR4) secretion measurements in lysates and supernatant were purchased from Elabscience (Houston, TX, USA), while the CellTiter 96^®^ cell proliferation kit was obtained from Promega Corporation (Madison, WI, USA). All the reagents were of high-grade purity.

#### 2.3.2. Cell Cultures

Human hepatic stellate cells (LX-2) were purchased from iXCells Biotechnologies (San Diego, CA, USA) and were used at 2 passages. Cell culture medium was DMEM supplemented with 5% Fetal Bovine Serum (FBS), antibiotics and antimycotics; all reagents were purchased from Sigma-Aldrich, Co. (Heidelberg, Germany).

#### 2.3.3. Cell Viability Assay

MTS assay (CellTiter 96^®^ AQueous Non-Radioactive Cell Proliferation Assay) was used to evaluate the cytotoxicity of the blueberry extract, NPCS and the hybrid compound (NPCS-VL), in accordance with the ISO 10993-12:2012 standards [25]. LX-2 cells were plated at 10^4^ per well in 96-well ELISA plates (TPP, Switzerland) and, after 24 h, they were exposed to different concentrations of VL, CS, NPCS and NPCS-VL. Subsequently, the cells were washed and then treated with 20 μL MTS/PMS per 100 μL medium for 2 h. Cell numbers were estimated by measuring their absorbance at 490 nm, using the ELISA plate reader (Tecan, Switzerland). IC50 levels were quantified for each substance. Additionally, the cell viability was assessed by measuring lactate dehydrogenase (LDH) activity in supernatant, using a spectrophotometric method. This parameter is a marker of membrane damage and quantifies cell lesions. Sodium pyrophosphate buffer in a concentration of 0.05 M, pH 8.8, containing 7.75 × 10^−2^ M lactate (final concentration), was mixed with 5.25 × 10^−3^ M NAD and then added to the samples. The mixture’s absorbance was measured at 340 nm. One unit of activity was defined as the amount required to catalyze the reduction of 1 mmol NAD/min mg protein [26].

#### 2.3.4. Cell Lysates

LX-2 cells were plated at 5 × 10^4^ cells/mL in 24-well plates and were cultured in growth medium supplemented with FBS 1%. Half of the cells were incubated with 10 ng/mL TGF-β for 24 h. Afterwards, the cells were treated with VL, NPCS and NPCS-VL. The cells were divided into eight groups as follows: I: untreated cells (negative control group); II: received only 10 ng/mL TGF-β; III: received only 25 μg/mL NPCS; IV: received 10 ng/mL TGF-β + 25 μg/mL NPCS; V: received only 25 μg/mL VL; VI: received 10 ng/mL TGF-β + 25 μg/mL VL; VII: received 25 μg/mL NPCS + 25 μg/mL VL; VIII: received 10 ng/mL TGF-β + 25 μg/mL NPCS + 25 μg/mL VL. All tests were performed in triplicate. The cell lysates used for the biochemical assays were obtained using the following method: the cells were collected from the surface of the culture, placed in ice, rinsed twice in the buffer solution and then lysed in 0.3 mL cold solution of 0.1% Triton X-100, 0.25 mol/L NaCl, 1 mmol/L EDTA, 50 mmol/L NaF, 1 mmol/L dithiothreitol, 0.1 mmol/L Na3VO4, 50 mmol/L TRIS HCl and protease-inhibitor complex 1% (Sigma). Lysates were centrifuged and protein concentration in cell culture supernatant was determined using the Bradford method, according to the manufacturer’s specifications (Bio-Rad Laboratories, Richmond, CA, USA).

#### 2.3.5. Oxidative Stress and Inflammation Markers Assessment

The oxidative stress generated by the administration of TGF-β alone or in combination with VL, NPCS, or NPCS-VL was assessed by measuring the malondialdehyde (MDA) as a parameter of lipid peroxidation. MDA levels were determined using the fluorimetric method with thiobarbituric acid, as previously described by Conti et al., slightly modified [27]. The sample was mixed with 10 mM thiobarbituric acid and 75 mM K_2_HPO_4_ and boiled in a water bath for one hour. Then, it was cooled and the adducts were extracted in n-butanol. MDA was measured by spectrophotometry and the result was expressed as nmoles/mg protein. The inflammation was assessed by ELISA measurement of IL-1β, IL-1α and TLR4 secretion in lysates and in supernatant for each of the study groups.

#### 2.3.6. Immunofluorescence Staining of Collagen I

Type I collagen expression was assessed through fluorescence microscopy using an antibody against collagen I conjugated with Alexa Fluor 455. Cells were cultured with each tested compound and their mixture for 24 h, followed by fixation with 2% paraformaldehyde and permeabilization with Triton X. Subsequently, the cells were blocked with BSA 5% for 1 h and then incubated with anti-mouse antibody against collagen I for an additional 1 h at room temperature. Nuclei were marked with DRAQ5 staining (Sigma Chemical Co., St. Louis, MO, USA). Microscopic images were captured using an Olympus CKX 41 microscope (Hamburg, Germany) at an original magnification of 20×.

#### 2.3.7. Fibrosis Evaluation

In order to quantify the fibrosis severity, Western blot analysis of α-SMA, type I collagen and type III collagen expression was used. For each study group, 20 μg of protein from the cell lysates were separated by electrophoresis using the Biorad Western Blot MiniProtean system on acrylamide 12.5% gels. Then, the gels were transferred onto polyvinylidene difluoride (PVDF) membranes. Blots were washed, blocked and incubated with specific primary antibodies anti-α-SMA, anti-collagen I and anti-collagen III, at a 1:500 dilution, for 45 min at 4 °C (Santa Cruz Biotechnology, USA). Then, the membrane was incubated overnight with specific secondary antibodies, in 1:1000 dilution (Santa Cruz Biotechnology). Chemiluminescence was used for the immunolocalisation, in accordance with the protocol (Sigma-Aldrich, St. Louis, MO, USA). GADPH was used as a loading control for protein normalization.

### 2.4. Statistical Analysis

The statistical significance of cell viability differences between the treated and control groups was evaluated with the two-way ANOVA and paired Student’s *t*-test, followed by Bonferroni posttest using GraphPad Prism version 5.00 for Windows (GraphPad Software, San Diego, CA, USA), www.graphpad.com (accessed on 5 May 2023). For biochemical analysis, the differences were assessed by one-way ANOVA and Tukey posttest. A *p* value < 0.05 was considered statistically significant. All reported data were expressed as the mean of triplicate measurements ± standard deviation (SD).

## 3. Results

### 3.1. Synthesis of Fruit Extracts and Gold Nanoparticles and Their Characterization

In order to characterize the obtained fruit extracts, the total phenolic content was chosen as an indicator. The total polyphenol content (TPC) in VL extract, determined by the Folin–Ciocalteu method, was found to be 174.23 ± 6.1 μg GAE/mL extract, while, for CS extract, the TPC value determined by the same method was 322.04 ± 11.7 μg GAE/mL extract. Due to its antioxidant activity, CS fruit extract can be successfully used to obtain NPCS by reducing Au^3+^ ions. These nanoparticles are also stabilized by the bioactive compounds of the extract, preventing their agglomeration. To confirm the formation of metallic gold, UV-Vis spectroscopy was used. The UV-Vis spectrum of the CS fruit extract (Figure 1a) shows a maximum at 504 nm, while, in the spectrum of the colloidal gold solution, a peak at λmax = 542 nm can be observed, which is the characteristic value of the metallic gold surface plasmonic resonance (Figure 1a) [28]. Transmission electron microscopy (TEM) was used to investigate the shape and morphological characteristics of NPCS resulting from the reduction of gold ions with bioactive compounds from the CS fruit extract. The TEM image (Figure 1b) illustrates the production of spherical gold nanoparticles, uncluttered, with an average diameter of 19 nm.

The hydrodynamic diameter of NPCS was 283.1 nm (Figure 2a), and their zeta potential was −13.1 mV (Figure 2b).

XRD analysis confirmed the formation of gold nanoparticles (the identification ICDD DB card number 00-001-1172). Three distinctive diffraction peaks at 2θ values (2-theta(deg)56.95(17), 64.6(6) and 77.4(4)) corresponding to the reflection planes of (111), (200), (220), characteristic to gold, were noticed (Figure 3a). In EDX analysis, the presence of gold nanoparticles was confirmed, as seen in Figure 3b. The spectra were obtained using an Oxford Instrument windowless detector and AZtec Software, as previously published [24]. The samples were prepared by dropping a few μL of diluted ethanol suspension of the sample on the copper grid (Figure 3b).

### 3.2. Biological Assays

#### 3.2.1. Cell Viability Assay

The effect of CS, VL, NPCS and NPCS-VL on LX-2 cells’ viability was assessed by MTS (Figure 4). The doses of both fruit extracts and NPCS, which maintained viability above 70%, were not considered toxic, whereas the doses which lowered it below 70% had toxicity. CS increased cell viability in a dose-dependent manner; administration of high doses of CS extract (100 and 200 μg/mL extract) showed the best viability, while none of the VL doses reduced cell viability below 70%. NPCS showed no toxicity, as all the administered doses maintained cell viability above 70%. A similar effect was noticed for the hybrid compound, with a slight decrease in cell viability when cells were exposed to 25 μg/mL NPCS + 40 μg/mL VL. The concentration of NPCS-VL used for the evaluation of its antioxidant, anti-inflammatory and antifibrotic effects was 25:25 μg/mL, in accordance with the results of the viability assay (Figure 4e).

In addition, LDH activity was measured in all studied groups. The enzyme activity was increased by TGF-β, illustrating the ability of TGF-β to induce membrane lesions. VL and NPCS-VL counteracted this effect, as LDH activities decreased after their administration (*p* < 0.05) (Figure 5).

#### 3.2.2. Oxidative Stress and Inflammatory Markers Assessment

Malondialdehyde (MDA), a marker of lipid peroxidation, was reduced by NPCS administration, demonstrating the free-radical scavenging effect of nanoparticles (*p* < 0.05) on cells preincubated with TGF-β (Figure 6a). NPCS-VL administration showed a slight, statistically insignificant decrease in MDA content, while VL amplified the lipid peroxidation of cells incubated with TGF-β. Toll-like receptor 4 (TLR4) expression increased in cells treated with TGF-β, while NPCS, VL and NPCS-VL were all efficient in lowering its concentration in the supernatant (Figure 6b) compared to cells exposed only to TGF-β (*p* < 0.001).

NPCS, VL and NPCS-VL had the ability to reduce IL-1β levels in cell lysates of LX-2 incubated with TGF-β (*p* < 0.05) compared to cells exposed to TGF-β (Figure 6c). Similarly, treatment with VL and NPCS-VL resulted in a decrease in IL-1α in the supernatant (*p* < 0.05) (Figure 6d).

#### 3.2.3. Immunofluorescence Staining of Collagen I

Fluorescence microscopy images revealed alterations of collagen I expression in cells incubated with TGF-β, resulting in an augmentation of its thickness. Administration of NPCS, VL and NPCS-VL mitigated collagen I synthesis (Figure 7).

#### 3.2.4. Fibrosis Markers Evaluation

TGF-β administration increased the expression of the alpha isoform of SMA by activated HSC (*p* < 0.05) (Figure 8b). The cells treated with VL, NPCS or NPCS-VL afterwards showed a downward trend in α-SMA expression, but it was statistically insignificant (Figure 8b).

The expression of type I and type III collagen was not significantly increased by TGF-β administration. Neither the natural extracts nor the hybrid compound were efficient in reducing the expression of these two types of collagen (Figure 8c,d). Moreover, the hybrid mixture enhanced the collagen III expression in cells exposed to TGF-β compared to TGF-β-treated cells (*p* < 0.05).

## 4. Discussion

The treatment of chronic liver disease remains a challenge for clinicians, as the currently approved drugs used for promoting the regression of liver fibrosis have limited effects. Phytodrugs have been extensively studied for their beneficial effects in this pathology. The results have shown that some of them (e.g., Curcumin, Silymarin) have strong hepatoprotective and antifibrotic properties, but their oral bioavailability is low. On the other hand, others (e.g., Resveratrol) have been found to reduce inflammatory markers in non-alcoholic fatty liver disease (NAFLD) patients, but they have insignificant antifibrotic properties [29].

Over the last years, nanotechnology has exhibited significant development in the treatment of liver fibrosis, as it facilitates combination therapy and targeted drug delivery. Many studies demonstrated that nanoparticles decrease levels of fibrosis marker in experimentally induced hepatic injury when administered in combination with Silymarin [30,31], and they reduce hepato-renal injury produced by acetaminophen in rats [32]. Given this evidence, the use of a hybrid material obtained by combining gold nanoparticles phytoreduced with natural compounds and a fruit extract known for its antioxidant and anti-inflammatory properties could be an effective therapeutic strategy in CLD. The potential of this mixture to improve liver fibrosis evolution and reduce disease progression was evaluated through an analysis of oxidative stress, inflammation and fibrosis markers.

In our experiment, none of the administered doses of CS, VL, NPCS and NPCS-VL showed toxicity, as they all maintained cell viability above 70%. NPCS and VL did not induce cell membrane damage, while their mixture increased LDH activity in comparison to the untreated study group. In LX-2 cells incubated with TGF-β, VL alone and NPCS-VL exhibited a protective effect against cell membrane lesions. IL-1β levels increased after TGF-β administration; this effect was attenuated by NPCS, VL and hybrid material administration, indicating the anti-inflammatory properties of the tested compounds. α-SMA showed a statistically insignificant downward trend in LX-2 cells treated with NPCS or NPCS-VL. VL, and especially NPCS-VL, amplified the expression of collagen III.

It is known that hepatic stellate cells are one of the four types of cells which form the liver structure and represent the main effectors of liver fibrogenesis. Their activation by various factors (TGF-β being among the most potent ones) results in an increased synthesis of ECM components, mainly fibrillary collagens. In its natural evolution, this process becomes irreversible and leads to cirrhosis, ultimately causing hepatic failure. Additionally, the stimulation of the TGF-β pathway induces reorganization of the actin cytoskeleton, generating rapid and long-term modifications of actin dynamics, changes in cell morphology and altered cell motility [33]. Actin polymerization at cell edges forms stable bundles known as stress fibers [34], a process mediated by the activation of the Rho GTPase family and Smad proteins [35]. The upregulation of actin genes and cytoskeleton remodeling are key factors in epithelial–mesenchymal transition (EMT), leading to alterations in cell shape and function and, consequently, contributing to tissue stiffness.

Understanding the underlying pathogenetic mechanisms of liver fibrosis is essential for the development of an efficient therapeutic agent. Hepatic stellate cell activation, inflammation, free radical production, alteration of cellular cytoskeleton and fibrosis are key processes responsible for the occurrence and perpetuation of liver disease. Hence, our study was performed on hepatic stellate cells exposed to TGF-β and treated with gold nanoparticles phytoreduced with CS extract, VL extract or their mixture.

The gold nanoparticles were obtained by green synthesis using CS extract, known for its high content of antioxidants and anti-inflammatory substances. This natural compound was selected based on previous studies that demonstrated the benefits of hybrid materials, which used extracts from Cornus species (e.g., *Cornus mas*) as capping agents on experimentally induced inflammation [36]. The UV-Vis spectrum, EDX and XRD analysis of NPCS confirmed the synthesis of spherical, stable gold nanoparticles with a mean diameter of 19 nm. The hydrodynamic diameter of NPCS was measured at 326 nm, demonstrating the binding of polyphenols to the surface of the gold nanoparticles.

Afterwards, NPCS were mixed with VL extract for the purpose of obtaining a hybrid compound with potentially high antioxidant and anti-inflammatory properties. It was demonstrated that blueberry fruit’s high content in polyphenols, both flavonoid (e.g., anthocyanins, procyanidins) and nonflavonoid types (e.g., chlorogenic acid), is responsible for these beneficial effects [37] and, in addition, for its antimutagenic and antitumor properties [20]. Moreover, VL also serves as a moderate source of ascorbic acid, which contributes to its free radical scavenging activity [38,39]. Studies have already highlighted the antioxidant and anti-inflammatory properties of blueberry extract in experimental models of liver fibrosis. Wang et al. found that blueberry reduced MDA levels in rats with CCl4-induced liver disease [40], while Yan et al. demonstrated that this natural substance attenuated liver fibrosis, protected the intestinal barrier and maintained gut microbiota homeostasis in a CCl4-induced toxic hepatitis model [41]. Our study demonstrated that VL alone and in combination with NPCS reduced inflammation markers, whereas NPCS-VL attenuated the oxidative stress.

In order to quantify compound toxicity, both cell viability and LDH activity were evaluated. Lactate dehydrogenase is a ubiquitous enzyme that catalyzes the reversible conversion of pyruvate to lactate. It is commonly used as an indicator of cytotoxicity, as LDH activity rises in response to tissue injury. Liver is no exception, as LDH (especially LDH-5 isoform) has significant expression in hepatocyte cytoplasm, and it is released in the event of membrane damage [42]. The MTS test revealed no toxicity of the tested compounds, and the results were confirmed by LDH activity, especially for VL and NPCS-VL.

Oxidative stress occurs as a result of the disproportion between the production of reactive oxygen species (ROS) and the ability of antioxidants to counteract their effects. Increased levels of free radicals play an important role in the pathogenesis of most etiologies of CLD and have been directly linked to fibrosis. They can activate quiescent HSC and induce their differentiation into myofibroblasts. This theory is supported by co-culture studies of HSC and Kupffer cells (KC), which showed that ROS produced by NADPH oxidase (highly expressed in KCs) increased collagen I production [43]. Moreover, antioxidant compounds, such as Silymarin, have been reported to possess antifibrotic effects [44]. TGF-β signaling, a pivotal pathway in liver fibrogenesis, can be activated by ROS either by inducing its expression or augmenting its release [45]. On the other hand, TGF-β stimulates the production of ROS, creating a vicious cycle which perpetuates the redox imbalance and HSC activation.

To assess the oxidative stress, malondialdehyde (MDA), an end-product of lipid peroxidation, was estimated. In the literature, increased levels of this marker are associated with liver disease in patients (e.g., NAFLD) [46] as well as with experimentally-induced liver fibrosis (e.g., acetaldehyde-stimulated LX-2 cells) [47]. Our study found high levels of MDA after TGF-β exposure; NPCS and NPCS-VL reduced the ROS formation triggered by TGF-β. Similar results were reported in an experimental model of liver injury induced by ethanol and methamphetamine after treatment with gold nanoparticles [17].

Inflammation is another pivotal hallmark of liver fibrosis. Inflammatory signals and immune cells activate HSC which, in turn, induce inflammatory pathways and subsequent cytokine secretion [48]. Cytokines are mediators of inflammation and act as markers of tissue injury and disease activity. They can be either pro- or anti-inflammatory, and an imbalance between these two types is characteristic for many diseases, including chronic hepatitis [49]. In CLD, the expression of pro-inflammatory cytokines is increased, with KCs being their major source. Upon activation by an injurious factor (e.g., alcohol, viral infection, dietary fats), KCs synthesize cytokines, induce ROS formation and destroy hepatocytes, with consequent release of DAMPs and other mediators [7,50]. All of these substances act as inducers of fibrogenesis by activating HSC, leading to excessive ECM production [29]. Moreover, cytokines can induce the disorganization of the actin cytoskeleton and the formation an increased amount of collagen fibers, promoting EMT and ECM alterations.

By modulating inflammation and oxidative stress, antifibrotic effects can be achieved. Our study demonstrated the pro-fibrogenic effect of TGF-β, a pleiotropic cytokine with pro-inflammatory activity. However, the antifibrotic properties of NPCS, VL and NPCS-VL could not be conclusively proven; they attenuated collagen I synthesis, yet the decrease in collagen I and α-SMA expressions was not statistically significant in Western blot. Our results are in accordance with data from the literature. For example, in an experimental model of CCl4-induced liver disease, the administration of Silymarin-conjugated gold nanoparticles reduced the proportion of α-SMA immunoreactive cells and the amount of collagen in liver tissue in comparison with the positive control group [51]. More in-depth studies are necessary to evaluate additional parameters and pathways involved in fibrogenesis and the dynamics of their action during treatment.

A distinctive aspect of our study was that NPCS-VL induced collagen III expression in LX-2 cells stimulated with TGF-β. It has been demonstrated that type I and type III collagen are the most abundant ECM proteins in the fibrotic liver [52]. Moreover, the N-terminal propeptide of procollagen type III (PIIINP) is used as a biomarker of liver fibrosis, reflecting collagen synthesis. During type III collagen synthesis, it is detached from procollagen III and released into the bloodstream [53]. NPCS reduced collagen III expression in cells stimulated with TGF-β, but, when mixed with VL, the synthesis of collagen III was amplified. Investigating the pathogenetic mechanisms of collagen III secretion in greater detail may provide insight into the upregulation observed following NPCS-VL administration.

## 5. Conclusions

Oxidative stress, inflammation and fibrogenesis are key mechanisms in the pathogenesis of CLD; by inhibiting these three processes, disease progression could be stopped and the ECM alterations could be reversed. Our results proved NPCS and NPCS-VL were particularly effective in reducing ROS production, while NPCS, VL and NPCS-VL all had anti-inflammatory properties and mitigated collagen I synthesis. Concerning fibrosis markers, the beneficial effects of the hybrid material are debatable, and further studies are needed to evaluate if NPCS-VL can genuinely modulate fibrogenesis.

## Figures and Tables

**Figure 1 biomolecules-13-01271-f001:**
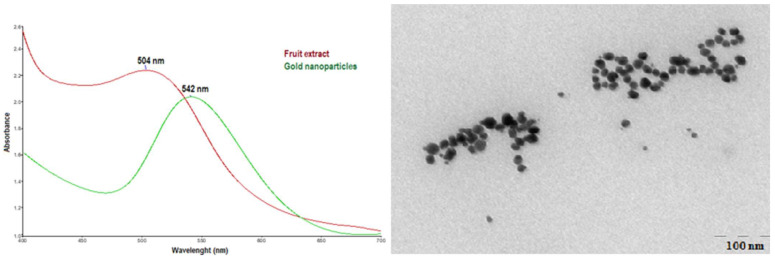
(**a**) UV-Vis spectra of VL extract and NPCS and (**b**) TEM images of NPCS.

**Figure 2 biomolecules-13-01271-f002:**
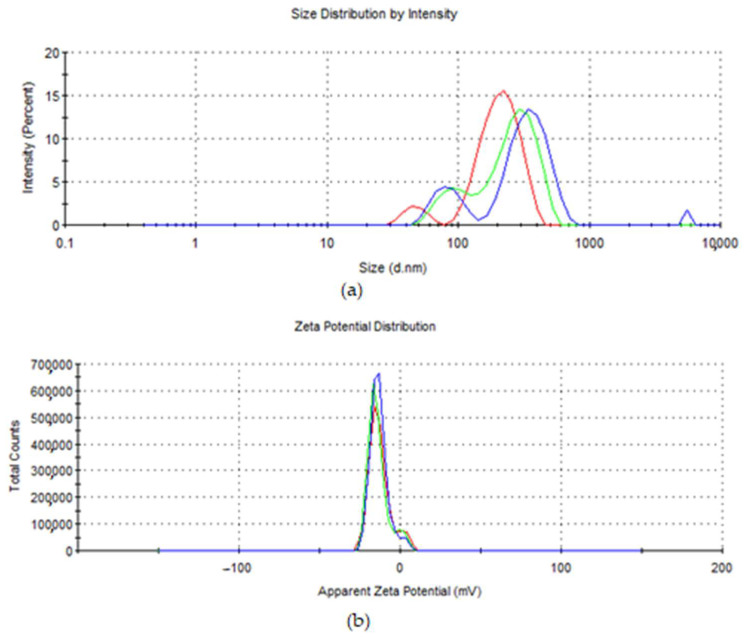
(**a**) The hydrodynamic diameter and (**b**) the zeta potential of NPCS.

**Figure 3 biomolecules-13-01271-f003:**
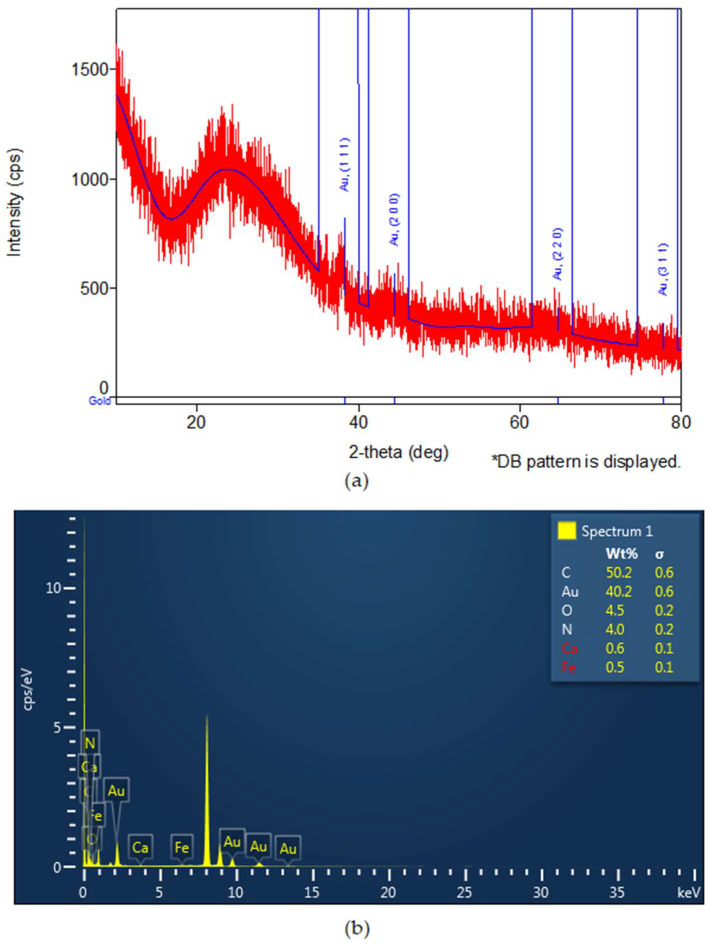
(**a**) XRD and (**b**) EDX analysis of NPCS.

**Figure 4 biomolecules-13-01271-f004:**
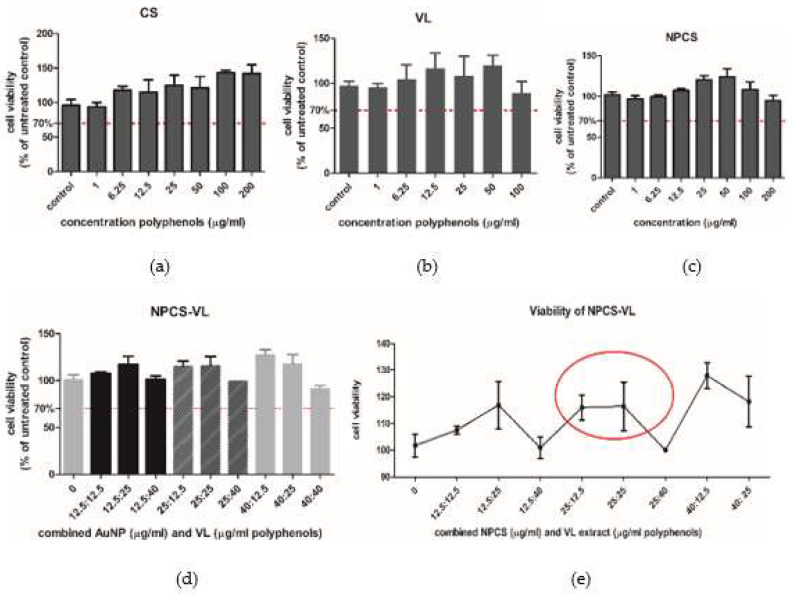
Viability of LX-2 cells exposed to different concentrations of CS extract (**a**), VL extract (**b**), NPCS (**c**) and NPCS-VL (**d**). The cells were treated with different concentrations (0–100 μg/mL) of CS, VL, NPCS and NPCS-VL, and their viability was compared to unexposed, control cells. The viabilities of NPCSVL (**e**) mixtures in variable concentrations (12.5:12.5; 12.5:25; 12.5:40; 25:12.5: 25:25; 25:40; 40:12.5; 40:25) were evaluated. For biochemical tests the NPCS-VL concentration used was 25:25 μg/mL. Data are presented as mean OD 540 nm ± SD, (n = 3).

**Figure 5 biomolecules-13-01271-f005:**
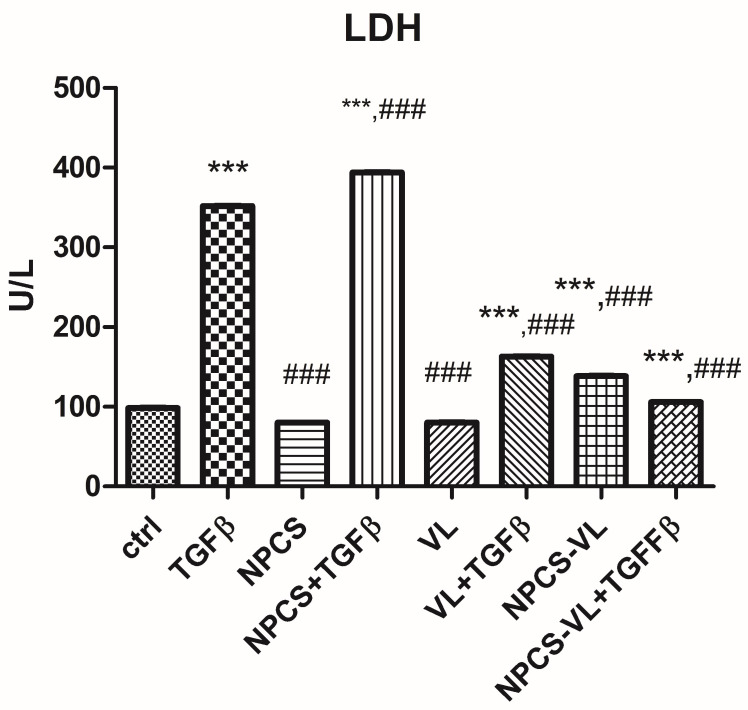
LDH activity in LX-2 cells exposed or not to TGF-β and treated with NPCS, VL and NPCS-VL. TGF-β increased membrane damage (*p* < 0.001), effect amplified by the association of NPCS (*p* < 0.001). VL and NPCS-VL significantly reduced the LDH activity of cells exposed to TGF-β (*p* < 0.001). Data are means ± standard deviation. The statistical significance of the difference between treated and control group was evaluated with one-way ANOVA, followed by Tukey’s posttest. *** *p* < 0.001 vs. control group; ### *p* < 0.001 vs. TGF-β exposed cells.

**Figure 6 biomolecules-13-01271-f006:**
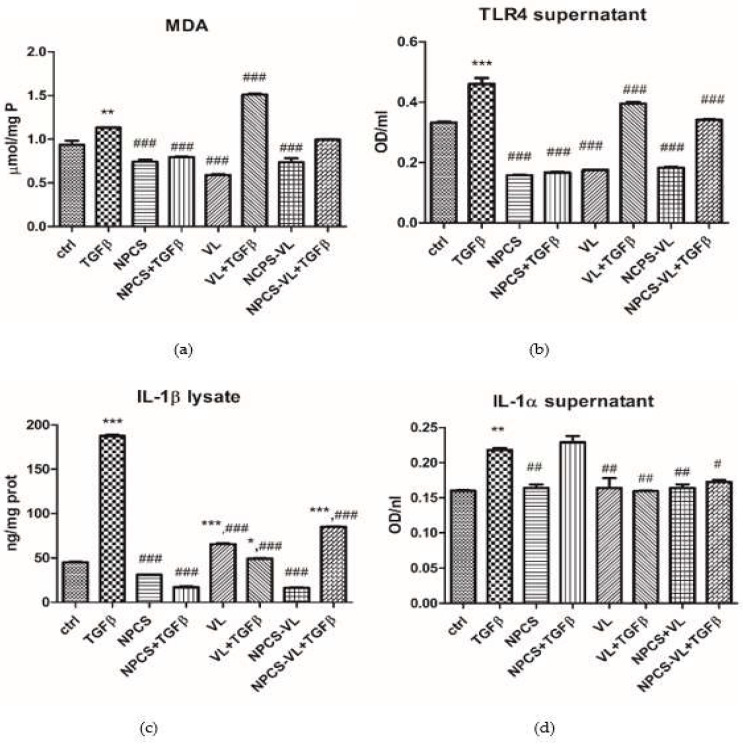
Oxidative stress and inflammatory marker assessment in LX-2 cells exposed to TGF-β and treated with NPCS, VL and NPCS-VL. (**a**) Malondialdehyde levels in cell lysates increased after TGF-β stimulation (*p* < 0.01), effect diminished by NPCS and NPCS-VL (*p* < 0.001). VL extract amplified lipid peroxidation after TGF-β exposure. (**b**) TRL4 levels in supernatant increased after TGF-β exposure (*p* < 0.001) and decreased in cells treated with NPCS, VL and NPCS-VL (*p* < 0.001). (**c**) IL-1β levels in cell lysates enhanced after TGF-β administration and were reduced after NPCS, VL and NPCS-VL treatments (*p* < 0.001). (**d**) IL-1α secretion in supernatant increased in cells exposed to TGF-β and diminished in cells treated with VL and NPCS-VL (*p* < 0.001). Data are means ± standard deviation. The statistical significance of the difference between treated and control group was evaluated with one-way ANOVA, followed by Tukey posttest, * *p* < 0.05, ** *p* < 0.01, *** *p* < 0.001 vs. control group and # *p* < 0.05, ## *p* < 0.01 and ### *p* < 0.001 vs. TGF-β exposed cells.

**Figure 7 biomolecules-13-01271-f007:**
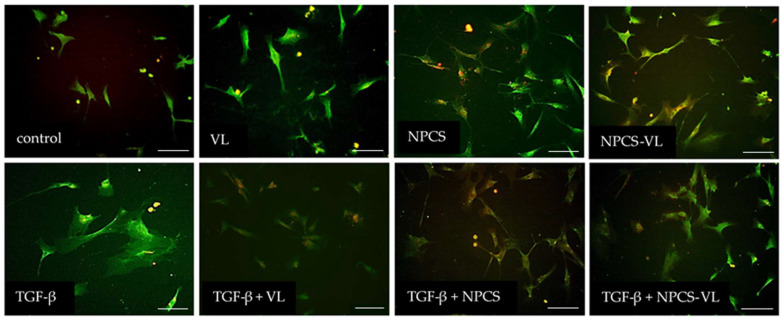
Fluorescence microscopy images of collagen I in LX-2 cells treated with NPCS, VL or NPCS-VL, with or without preincubation with TGF-β. Collagen I immunostaining displayed green fluorescence through the use of an antibody targeting collagen I conjugated with Alexa Fluor 455, while nuclei showed red fluorescence due to the DRAQ5 (646 nm/681 nm) staining. Control cells and those treated with VL exhibited normal collagen I expression. Conversely, cells incubated with TGF-β displayed thickened collagen fibers, indicating enhanced production. Administration of NPCS, VL and NPCS-VL yielded an improvement in collagen I synthesis, restoring collagen I fibers to their normal number and size. Scale bar represents 20 μm.

**Figure 8 biomolecules-13-01271-f008:**
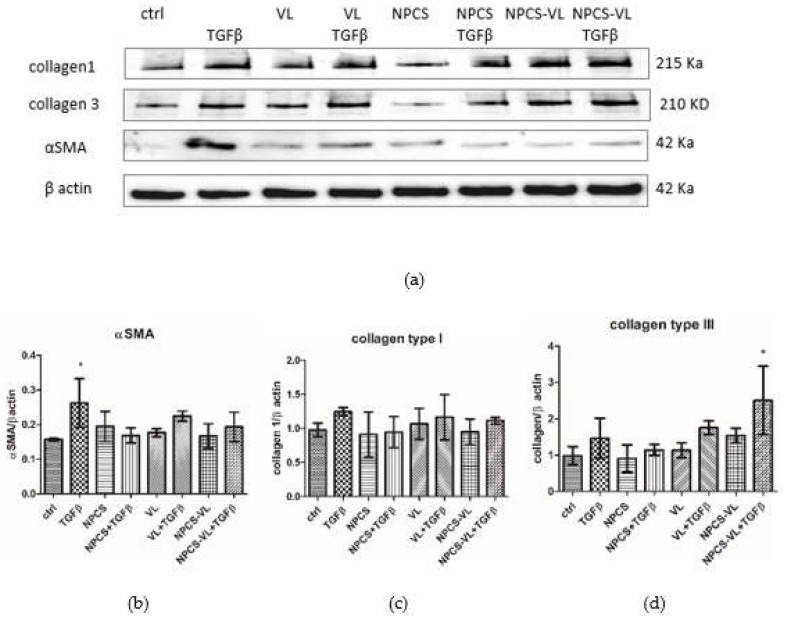
Fibrosis marker expression, α-SMA, collagen I and III, in LX-2 cells exposed to TGF-β. LX-2 cells exposed to TGF-β were treated with NPCS, VL and NPCS-VL. (**a**) Image analysis of Western blot bands was performed by densitometry (upper panels), and the results normalized to β-actin are shown in the lower panels (**b**–**d**). α-SMA expression increased in cells preincubated with TGF-β (*p* < 0.05), while the treatment with NPCS-VL enhanced the expression of collagen III (*p* < 0.05). Data are means ± standard deviation. The statistical significance of the difference between the treated and control groups was evaluated with one-way ANOVA, followed by Tukey posttest, * *p* < 0.05 vs. control group.

## Data Availability

Not applicable.

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
