# Peer review of "Hybrid Material Based on Vaccinium myrtillus L. Extract and Gold Nanoparticles Reduces Oxidative Stress and Inflammation in Hepatic Stellate Cells Exposed to TGF-β"

_biomolecules, 2023, doi:10.3390/biom13081271_

Round 1

Reviewer 1 Report

Review report on the:

Journal Biomolecules (ISSN 2218-273X), Manuscript ID biomolecules-2539509

Title: Hybrid material based on Vaccinium L. extract and gold nanoparticles reduces oxidative stress and inflammation in hepatic stellate cells exposed to TGF-β

The presented manuscript is generally well-written, the structure is clear and it is interesting to the reader. The authors presented a method to obtain gold nanoparticles by green synthesis, which is recently a very popular topic in nanotechnology. The advantages of green synthesis include the reduction of harsh chemicals usage, as well as the possibility of including other beneficial molecules extracted from the plants into the nanoparticles-plant extract mix. However, the results obtained from this type of study have to be evaluated carefully, because the molecules present in the extract might have a significant impact on the outcome. The authors managed to present most of the necessary controls.

I have some concerns related to the metabolic cytotoxicity assays and the nanoparticles colloids preparation:

11)    Please explain the MTS and LDH assay approach: why did you wait another 24 hours after cell washing to perform the assay? According to what you have written: cells were exposed to the NPs and plant extracts for 24h, then you washed them and waited another 24 hours. This way you created a window for cells to proliferate in the absence of the treatment agents. Even though they might have had cytotoxic or antiproliferative effects, it is difficult to evaluate now. The results after another 24h might be significantly biased.

22)   The results obtained by metabolic assays might be non-conclusive in the case of nanoparticles and nanomaterials studies (non-specific interactions between NPs and reagents, even when NPs are trapped within cells or might be stuck on the cell culture flask/dish bottom etc.). Thereby, it is advised to control the experiments also by visual confirmation on a microscopic level. I would strongly recommend presenting the results alongside some images from microscopy (e.g. at least live imaging after treatment).

3)  In section 2.2. Synthesis of NPCS nanoparticles and their characterization I did not find any information on how the stock solution of synthesized, centrifuged and washed gold NPs was prepared. There is a lack of information on what basis the authors calculated (and prepared) the concentrations for the follow-up tests.

The most important issue:

4.       All the images in the file are of very low quality (blurred), please provide better resolution images, some of them also contain some errors and unnecessary underlines in the text (e.g. Fig.6)

Otherwise, I find the research valuable and important for the field. I recommend the manuscript be revised in the above-mentioned aspects before publication.

Author Response

Reviewer 1 Comments and Suggestions for Authors

Review report on the: Journal Biomolecules (ISSN 2218-273X), Manuscript ID biomolecules-2539509

Title: Hybrid material based on Vaccinium L. extract and gold nanoparticles reduces oxidative stress and inflammation in hepatic stellate cells exposed to TGF-β

The presented manuscript is generally well-written, the structure is clear and it is interesting to the reader. The authors presented a method to obtain gold nanoparticles by green synthesis, which is recently a very popular topic in nanotechnology. The advantages of green synthesis include the reduction of harsh chemicals usage, as well as the possibility of including other beneficial molecules extracted from the plants into the nanoparticles-plant extract mix. However, the results obtained from this type of study have to be evaluated carefully, because the molecules present in the extract might have a significant impact on the outcome. The authors managed to present most of the necessary controls.

I have some concerns related to the metabolic cytotoxicity assays and the nanoparticles colloids preparation:

  1.  Please explain the MTS and LDH assay approach: why did you wait another 24 hours after cell washing to perform the assay? According to what you have written: cells were exposed to the NPs and plant extracts for 24h, then you washed them and waited another 24 hours. This way you created a window for cells to proliferate in the absence of the treatment agents. Even though they might have had cytotoxic or antiproliferative effects, it is difficult to evaluate now. The results after another 24h might be significantly biased.

Response: We agree with the reviewer, the protocol did not include a 24h wait, this was a mistake and we corrected it. The cells were treated with nanoparticles for 24h, then medium was collected for the LDH assay.  Also, after nanoparticles exposure for 24h, the cells were washed and subjected to the MTS assay in fresh medium.

  1.  The results obtained by metabolic assays might be non-conclusive in the case of nanoparticles and nanomaterials studies (non-specific interactions between NPs and reagents, even when NPs are trapped within cells or might be stuck on the cell culture flask/dish bottom etc.). Thereby, it is advised to control the experiments also by visual confirmation on a microscopic level. I would strongly recommend presenting the results alongside some images from microscopy (e.g. at least live imaging after treatment).

Response: All cell groups displayed a similar appearance in optical microscopy images, regardless of the administered substance. Because of this, we decided to evaluate their aspect using fluorescence microscopy with antibody conjugated with Alexa Fluor 455 against collagen I. For this purpose, the cells were cultured with each tested compound and their mixture for 24 h, followed by fixation with 2% paraformaldehyde and permeabilization with Triton X. Afterwards, the cells were blocked with BSA 5% for 1 hour and incubated with antimouse antibody against collagen I for another 1 hour, at room temperature. Nuclei were marked with DRAQ5 staining (Sigma Chemical Co., St. Louis, MO, U.S.A). Microscopic images were captured using an Olympus CKX 41 microscope (Hamburg, Germany) at an original magnification of 20×.

The results indicated that TGF-β administration induced alterations in the architecture of the cytoskeleton, increasing the secretion and thickness of collagen I. NPCS, VL and NPCS-VL administration mitigated collagen I synthesis and reduced the deposition of collagen fibres.

  1. In section 2.2. Synthesis of NPCS nanoparticles and their characterizationI did not find any information on how the stock solution of synthesized, centrifuged and washed gold NPs was prepared. There is a lack of information on what basis the authors calculated (and prepared) the concentrations for the follow-up tests.

Response: The resulting gold pellet was resuspended in 5 mL distilled water and the gold content of the obtained colloidal solution was evaluated by atomic absorption spectrometry using an AVANTA PM atomic absorption spectrometer (GBC Scientific Equipment, Braeside, Australia) equipped with a graphite furnace (GF 3000) and programmable auto-sampler (PAL 3000, GBC Scientific Equipment).

The information was added to the manuscript.

The most important issue:

  1. All the images in the file are of very low quality (blurred), please provide better resolution images, some of them also contain some errors and unnecessary underlines in the text (e.g. Fig.6)

Response: The figures were replaced with ones with higher resolution. The errors were corrected.

Otherwise, I find the research valuable and important for the field. I recommend the manuscript be revised in the above-mentioned aspects before publication.

Reviewer 2 Report

In this research, the authors prepared a hybrid material based on Vaccinium L. extract and gold nanoparticles and systematically demonstrated its potential of reducing reduces oxidative stress and inflammation in hepatic stellate cells in vitro. The authors intelligently leveraged by the differential function of the components and integrated them into a hybrid material and verified that the hybrid material realized the improved cell membrane damage and inflammation triggered by TGF-β, but increased the expression of collagen III, suggestive for a fibrogenetic effect of the hybrid material. The research is well designed, organized and presented, and should be published in Biomolecules without more revision, excepting adding more discussion about NPCS in the discussion section.

The function of NPCS mainly comes from the phytoreduced agents and capped agents, polyphenols from Cornus sanguinea L. Would the mixed extract solution from Cornus sanguinea and Vaccinum L realize a similar function to NPCS+VL? Another, gold nanoparticles may realize the delivery of extract from Cornus sanguinea, but few functions on the extract from Vaccinum L. What is the largest benefit (or function) of the gold nanoparticle in the hybrid material?

Minor editing of English language required

Author Response

Reviewer 2

Comments and Suggestions for Authors

In this research, the authors prepared a hybrid material based on Vaccinium L. extract and gold nanoparticles and systematically demonstrated its potential of reducing reduces oxidative stress and inflammation in hepatic stellate cells in vitro. The authors intelligently leveraged by the differential function of the components and integrated them into a hybrid material and verified that the hybrid material realized the improved cell membrane damage and inflammation triggered by TGF-β, but increased the expression of collagen III, suggestive for a fibrogenetic effect of the hybrid material. The research is well designed, organized and presented, and should be published in Biomolecules without more revision, excepting adding more discussion about NPCS in the discussion section.

The function of NPCS mainly comes from the phytoreduced agents and capped agents, polyphenols from Cornus sanguinea L. Would the mixed extract solution from Cornus sanguinea and Vaccinum L realize a similar function to NPCS+VL? Another, gold nanoparticles may realize the delivery of extract from Cornus sanguinea, but few functions on the extract from Vaccinum L. What is the largest benefit (or function) of the gold nanoparticle in the hybrid material?

Response: We cannot affirm that the mixed extracts obtained from Cornus sanguinea and Vaccinium myrtillus fruits realize a similar function to the mixture NPCS and VL extract as the combined effects of the two fruit extracts was not investigated. However, it’s well documented in our previous studies that gold nanoparticles functionalized with natural compounds from Cornus species have a strong ability to reduce oxidative stress and inflammation. That’s why we decided to investigate the combined, eventually even synergistic effects of the hybrid material based on gold nanoparticles bioconjugated with polyphenols from Cornus sanguinea and the antioxidant extract of Vaccinium myrtillus fruits.

Gold nanoparticles were chosen not only for their ability to deliver drugs, but also for their therapeutic properties. For example, their free radical scavenging properties on methamphetamine and ethanol-induced liver fibrosis was demonstrated (de Carvalho, T.G.; Garcia, V.B.; de Araújo, A.A.; da Silva Gasparotto, L.H.; Silva, H.; Guerra, G.C.B.; de Castro Miguel, E.; de Carvalho Leitão, R.F.; da Silva Costa, D.V.; Cruz, L.J.; Chan, A.B.; de Araújo Júnior, R.F. Spherical neutral gold nanoparticles improve anti-inflammatory response, oxidative stress and fibrosis in alcohol-methamphetamine-induced liver injury in rats. Int J Pharm. 2018, 548, 1-14). However, gold nanoparticles need to be capped with reducing agents in order to be administered. Multiple natural compounds were used to obtain gold nanoparticles by green synthesis. Among these, silymarin-coated (Abdullah, A.S.; El Sayed, I.E.T.; El-Torgoman, A.M.A.; Alghamdi, N.A.; Ullah, S.; Wageh, S.; Kamel, M.A. Preparation and Characterization of Silymarin-Conjugated Gold Nanoparticles with Enhanced Anti-Fibrotic Therapeutic Effects against Hepatic Fibrosis in Rats: Role of MicroRNAs as Molecular Targets. Biomedicines. 2021, 9, 1767) and curcumin-coated (Adlia, A.; Tomagola, I.; Damayanti, S.; Mulya, A.; Rachmawati, H. Antifibrotic Activity and In Ovo Toxicity Study of Liver-Targeted Curcumin-Gold Nanoparticle. Sci Pharm. 2018, 86, E41) gold nanoparticles showed antifibrotic properties. We chose Cornus Sanguinea extract for this purpose due to its high content in polyphenols, with proven anti-inflammatory and antioxidant properties.

Round 2

Reviewer 1 Report

The manuscript has been significantly improved and most of the issues has been address. Remaining minor issues:

- no scale bar on new Figure 7

- images are still low quality in PDF file (maybe it is the fault of the submission system, please resolve the issue with an editor)

Author Response

The manuscript has been significantly improved and most of the issues has been address. Remaining minor issues:

- no scale bar on new Figure 7

Response: We added a scale bar on Figure 7. We attached it to both the manuscript and the Word file.

- images are still low quality in PDF file (maybe it is the fault of the submission system, please resolve the issue with an editor)

Response: The quality of the images was improved. All graphs were exported from GraphPad at a resolution of 1200 dpi, but the quality probably diminished when they were assembled. We have sent all the images separately to the editor in order to ensure they have the highest possible resolution.
